# Adjuvants for COVID-19 Vaccines

**DOI:** 10.3390/vaccines11050902

**Published:** 2023-04-26

**Authors:** Javier Castrodeza-Sanz, Iván Sanz-Muñoz, Jose M. Eiros

**Affiliations:** 1National Influenza Centre, 47005 Valladolid, Spain; jjcastrodeza@saludcastillayleon.es (J.C.-S.); jmeirosbouza@gmail.com (J.M.E.); 2Preventive Medicine and Public Health Unit, Hospital Clínico Universitario de Valladolid, 47003 Valladolid, Spain; 3Instituto de Estudios de Ciencias de la Salud de Castilla y León, ICSCYL, 42002 Soria, Spain; 4Microbiology Unit, Hospital Clínico Universitario de Valladolid, 47003 Valladolid, Spain; 5Microbiology Unit, Hospital Universitario Río Hortega, 47013 Valladolid, Spain

**Keywords:** vaccines, adjuvants, COVID-19, immune response

## Abstract

In recent decades, the improvement of traditional vaccines has meant that we have moved from inactivated whole virus vaccines, which provoke a moderate immune response but notable adverse effects, to much more processed vaccines such as protein subunit vaccines, which despite being less immunogenic have better tolerability profiles. This reduction in immunogenicity is detrimental to the prevention of people at risk. For this reason, adjuvants are a good solution to improve the immunogenicity of this type of vaccine, with much better tolerability profiles and a low prevalence of side effects. During the COVID-19 pandemic, vaccination focused on mRNA-type and viral vector vaccines. However, during the years 2022 and 2023, the first protein-based vaccines began to be approved. Adjuvanted vaccines are capable of inducing potent responses, not only humoral but also cellular, in populations whose immune systems are weak or do not respond properly, such as the elderly. Therefore, this type of vaccine should complete the portfolio of existing vaccines, and could help to complete vaccination against COVID-19 worldwide now and over the coming years. In this review we analyze the advantages and disadvantages of adjuvants, as well as their use in current and future vaccines against COVID-19.

## 1. Introduction

Modern vaccinology is a field in continuous evolution that is constantly subjected to the appearance of new challenges, which must be solved in the shortest possible time to transfer new improvements to the population. This was the case in the start of 2019 of the COVID-19 pandemic caused by SARS-CoV-2, which forced the scientific community to work in unison to design, test and administer new vaccines against this new pathogen in record time. In fact, since the appearance of the first cases of SARS-CoV-2 in December 2019 [1], only three months passed until the first clinical trials with the first vaccines against COVID-19 began, and one year until they began to be administered in people living in the European Union.

After three years of the pandemic, the number of vaccine designs against COVID-19, both approved and in clinical, pre-clinical and development trials, amounts to more than 370 [2]. The fact that, despite that great steps have been taken in the prevention of COVID-19, new vaccines against this disease continue to be so actively sought, shows that there is still a long way to go to, among other factors, protect the entire world population, democratize and make vaccines more accessible, adapt them to new variants of COVID-19 that appear in the future and, of course, make them more effective and to produce longer lasting protection.

Before the COVID-19 pandemic, there were already multiple vaccine designs against different microorganisms. Depending on the pathogen, inactivated vaccines are used if the microorganism is dead and not capable of producing infection [3], or, on the other hand, attenuated ones, if the microorganism has been subjected to different processes transforming it into a virulent, but leaving intact the capacity to replicate in the vaccinated host [4]. There are also vaccines that use only parts of the microorganism, such as vaccines based on antigen subunits or those that use toxoids. Additionally, and given that in certain groups of people a reinforcement of immunity is needed, there are also vaccines designed for increasing immunogenicity which, through different mechanisms, help to increase the vaccination response in those people who need it, mostly risk groups [5,6]. One of the vaccine designs that has the greatest interest in this aspect is one that includes adjuvants in its formulation. These vaccines have multiple advantages since they allow the immune response induced by the vaccine to be potentiated in different terms, either at the humoral or cellular level, being especially relevant in the elderly, or those whose ability to fight against infections is diminished.

Until the emergence of new designs in vaccinology during the pandemic, one of the main problems that has arisen with the evolution of vaccines since their origins is that, as the antigens became simpler and more purified, their immunostimulant capacity decreased in parallel. Many of the older inactivated vaccines were made up of the whole organism, which offered a moderate vaccine response (immunogenicity), but low tolerability due to frequent occurrence of side effects [7]. To avoid this reactogenicity, vaccines progressively evolved to designs that contained only parts of the microorganism, subunits, or even purified antigens [8]. However, this caused the immunogenicity to slowly decline compared to whole virus vaccines, which, for some risk groups such as people over 65 and those who are immunocompromised, is not acceptable. Due to their nature, purified protein antigen vaccines without adjuvants induce a modest antibody response with little or no T cell response [9]. The appearance of adjuvants made it possible to recover the immunogenicity of these kinds of vaccines, also demonstrating, in most cases, much better tolerability profiles than those of traditional whole inactivated organism vaccines [8] (Figure 1).

Furthermore, inactivated vaccines require large and multiple doses to confer protective immunity [10] and, in contrast with live-attenuated vaccines, elicit primarily humoral immunity, with little to no induction of cell-mediated immunity [11]. However, these inactivated vaccines also act as their own adjuvant, as they elicit a robust protective immune response compared to subunit vaccines [11]. Some of the inactivated vaccines that traditionally had incorporated adjuvants are, for example, poliomyelitis, hepatitis A, Japanese encephalitis virus and tick-Borne encephalitis, most of them of the alum type [12].

The first adjuvants that were used were aluminum salts, which began to be used in vaccines against Pertussis, Tetanus and Diphtheria in the 1920s, and are currently used for some vaccine designs against COVID-19 in different development phases of clinical trials [13]. The rest of the adjuvant formulas appeared later from the 1990s and 2000s, such as virosomes, some adjuvants such as MF59 (Novartis), AS01, AS03, AS04 (GSK) and different water–oil emulsions.

The acceleration in vaccinology brought about by the COVID-19 pandemic has led to the emergence of novel vaccination systems that have not been widely used before. However, many of the designs currently under development, and even some already on the market, use traditional technologies that are truly useful for preventing SARS-CoV-2 infection, such as vaccines that contain adjuvants. In a continuously changing horizon in which we are going to have to face the continuous appearance of new variants, it is necessary to explore different vaccine designs that help us keep the population protected. The objective of this review is to describe the main advantages of the use of adjuvants in future vaccines against COVID-19, as well as to show the current vaccine designs that use this technology and what their future may be in the short to medium term for prevention of this disease.

## 2. Advantages and Reasons for the Use of Adjuvants in Vaccines against COVID-19

Adjuvants have been widely used in vaccines against many human infectious diseases, demonstrating their great utility in increasing the vaccine response. These vaccines have shown various advantages that must be considered for the design of future vaccines against COVID-19. The main reasons for the use of adjuvants are the following, but also, some of them have limitations that are worth comment [14,15].

A.
They increase the response to the vaccine in the general population, and especially in those risk groups that show a reduced response due to age or different diseases:


Vaccines do not generate homogeneous protection in the population because there are multiple factors that condition the response to them. The most common are age, the existence of different diseases that can impact immunity, or the use of immunosuppressive medication, among others. These factors limit the response to old and new vaccines, directly impacting the ability to counteract an infection despite being vaccinated. Adjuvants are capable of increasing the response compared to antigen alone vaccines both in high-risk individuals and in the general population through, for example, an increase in seroconversion, an increase in the mean titers of antibodies detected, or stimulation of the cellular response [16,17,18]. Additionally, some works have also shown that vaccines with adjuvants are not only capable of increasing the amount of antibodies generated, but that this increase occurs specifically in those that have greater affinity for the antigens against which they are directed [19,20]. However, there are different types of adjuvants, and not all induce an optimal protection, so precision adjuvants may be used to avoid this issue.

B.
They make it possible to reduce the amount of antigen needed for each vaccine dose:


One of the main advantages of using adjuvants is that, due to their immunostimulation-enhancing effect, a smaller amount of antigen is required to produce a vaccine dose [9,21,22,23]. This makes it possible to manufacture a greater number of doses and, therefore, to vaccinate a larger population in less time, which is especially important in situations of increased demand such as a pandemic. During the first months of the COVID-19 pandemic, the main need was to vaccinate as many people as possible, and for this it was necessary to increase the number of doses produced. Using adjuvants for protein COVID-19 vaccines may have been useful during the first months of pandemic. However, some manufacturing issues made the production of this kind of adjuvants longer in this emergency, so prioritization of other vaccine platforms may be more suitable. However, this is a topic that may be revised for the next pandemic, in order to be ready to develop adjuvanted vaccines using an antigenic ready-to-use platform.

C.
They allow immunization with a smaller number of vaccine doses:


Another of the disadvantages of the use of traditional vaccines, and some of the current ones such as RNAm, is that they require multiple doses to achieve optimal responses [14]. At a logistical level, it is much more cost-effective for a person to acquire protection with the smallest number of doses possible, since, if they must get vaccinated several times, it supposes or implies different logistical challenges, in addition to the possibility of not attending future appointments. This can be detrimental because it implies a deficient vaccination, and does not ensure complete protection in the individual. Adjuvants allow the number of doses a person must be given to achieve protection to be reduced [21,22,24], which is also advantageous for faster immunization during a pandemic. The addition, for example, of the AS04 adjuvant to the hepatitis B antigen in the Fendrix (GSK) vaccine allowed a reduction from three to two doses in the vaccination regimen [25,26]. In the case of the emergence of a new virus, although adjuvants may require less doses than other vaccines, the population may need more than one vaccine dose to reach optimal protection. For that, the advantage of adjuvanted vaccines may be the need for fewer doses than other platforms for a more durable protection.

D.
They stimulate the immune response beyond antibodies:


As previously mentioned, generally the vaccine response to purified protein antigens is moderate/low in the production of antibodies, but very limited or absent in the stimulation of the T cell response. With the development of new adjuvants, we can achieve vaccines which can stimulate the helper T cell response by optimizing the quality and durability of the antibody response, as well as the induction of effector CD4 and CD8 cells to clear intracellular pathogens [9]. Indeed, for some intracellular antigens, activation of CD8+ T cells is essential because cytotoxic functions can help restrict infection and progression of these types of diseases [27]. However, not all the adjuvants are able to induce broad T-cell responses to the same extent. For example, there are some adjuvants such as ISCOMATRIX (Toll-like receptor 3 agonist) [28], some saponins derived from the tree *Quillaja brasiliensis* [29] and the Matrix-M adjuvant [30], that can stimulate both the CD8 and CD4 response [31]. However, there are other adjuvants that only stimulate one part of the cell-mediated immunity, such as alum-based adjuvants, which preferentially stimulate CD4 cells [32].

Moreover, the T-cell-stimulated response depends not only on the type of adjuvant, but also on the type of antigen and the carrier of this antigen [16]. For example, the use of liposomes and nanoparticles can stimulate CD8 and CD4 cells depending on their specific composition, causing the antigen to be processed by pathways located in the cytosol rather than the lysosomes, resulting in MHC class I presentation [33,34]. Antigen load is also a critical issue for the robustness of the CD8 response, as a higher antigen load trigger better responses [35].

E.
They stimulate heterotypic responses to different antigens:


Another of the advantages of adjuvants is that they are capable of producing heterotypic responses against antigens that have a phylogenetic relationship with the vaccine antigen, either due to the appearance of escape mutants, minor variants of a microorganism, etc. This has been previously demonstrated against influenza and HPV [36,37,38], and is especially important against SARS-CoV-2, since the continued evolution of the virus during the pandemic with the appearance of new variants has shown that the vaccines must be prepared to adapt easily and quickly to the variability of the virus.

## 3. Common Adjuvants Used in Non-COVID-19 Vaccines

As previously mentioned, the path of adjuvants has been a long one since their first uses in the 1920s. Currently, a century has passed since the first designs based on aluminum salts, and since then different types of adjuvants of higher or lower efficacy and quality have appeared, and have been used in multiple commonly used vaccines. Next, there will be a review of the main adjuvants and their use in current vaccines, as well as the mechanism by which they perform their function [39,40].

▪
Aluminum salts:


Aluminum salts, also known as alum or alum salts, were the first adjuvants used in vaccines. They were the compounds with which vaccines against tetanus, pertussis and diphtheria were formulated in the 1920s. They have a very high safety profile and this has meant that, while also being very cheap, they have been widely used ever since. In most cases, the side effects that are observed with aluminum salts are of a local type, such as inflammation at the injection site, with limited cases in which systemic effects are observed [41].

Aluminum salts have the function of stimulating responses, especially of the Th2 type [41,42], related to the stimulation of B lymphocytes for the production of antibodies. One of the main problems of some antigens is that they are soluble, so they are not easily recognized by the immune system [43]. For this, one of the main mechanisms of action of aluminum salts as an adjuvant is the creation of antigen deposits [44], which facilitates phagocytosis in addition to slowing its diffusion and release from the injection site, allowing the accumulation of cells to which the antigen can be presented [41]. This mechanism ensures a constant stimulation of the immune response, allowing a high antibody production. The antigen is absorbed into alum due to strong electrostatic interactions, resulting in the enhanced uptake of antigen by APCs. However, there are some recent studies that show that the formation of depots does not impact the stimulation of the immune system directly or it is not essential, so this pathway seems to be controversial [45].

Additionally, aluminum salts activate different kind of cells to the site of injection [46]. The alum adjuvants can recruit cells by themselves, or can induce the secretion of cytokines and chemokines for assisting this recruitment of cells [47]. Similarly, alum adjuvants attract dendritic cells and macrophages by different mechanisms, improving the presentation of antigens to other immune cells [43]. However, the main drawback of this kind of activation is that it is, in general, poor [48]. For that reason, sometimes alum adjuvants are used in combination with other salts or other adjuvants of a different nature to enhance the response.

Presently, there are more than 25 approved vaccines that contain some form of aluminum salts, alone or in combination, as an adjuvant. Below are some of the most commonly used adjuvants and in which vaccines they are included:
Aluminum hydroxide:
◦Inactivated hepatitis A◦Meningococcal group B◦Anthrax◦Diphtheria and tetanus (reduced antigenic content)◦Recombinant hepatitis B◦Hexavalent against diphtheria, tetanus, pertussis (acellular), hepatitis B (recombinant), poliomyelitis (inactivated) and Haemophilus influenzae type B◦Tetanus◦Japanese encephalitis◦Meningococcal type C◦Central European encephalitisAluminum phosphate
◦Tetanus toxoid, diphtheria toxoid, acellular pertussis◦Pentavalent against diphtheria, tetanus, pertussis (acellular), poliomyelitis (inactivated) and Haemophilus influenzae type B (conjugated)◦Pneumococcus◦Meningococcal type B (recombinant)Amorphous aluminum hydroxyphosphate sulfate:
◦Quadrivalent and 9-valent HPV◦Haemophilus influenzae type B◦Hepatitis B (recombinant)◦Combining two aluminum salts:◦Diphtheria, tetanus and acellular pertussis◦Pentavalent against diphtheria, tetanus, pertussis (acellular), hepatitis B (recombinant), poliomyelitis (inactivated)

▪
Emulsion adjuvants:


Emulsion adjuvants work in a similar way to aluminum salts, producing deposits of the antigen, in such a way that it is more accessible to the immune system [49]. These types of adjuvants have been used for more than two decades in more than 30 countries due to their safety, ease of manufacture and application, and notable effects in stimulating the vaccine response [50]. In fact, the use of biodegradable and biocompatible oils since the 1990s has accelerated the use of these compositions due to the great increase in the tolerability and safety of their use [51].

The main characteristic of this type of adjuvant is that, after administration, they produce what some authors refer to as a local “immunocompetence state”, which subsequently triggers a strong and long-lasting response in the lymph nodes. The induced response is for both Th1 and Th2, in addition to plasma cells, memory B lymphocytes, and high titers of neutralizing antibodies, which are even capable of carrying out heterotypic responses [21,37,52,53]. However, the effect of these adjuvants does not stop at bioaccumulation, but also produces antigen uptake through monocytes, macrophages, and dendritic cells, which secrete different cytokines that recruit more cells to the injection site. Due to these characteristics, there are currently several designs of vaccines against COVID-19 that are exploring this type of adjuvant [54,55].

The two main emulsion adjuvants currently in use are MF59 and AS03. Both are made up of an emulsion based on squalene, which is an oil from shark liver, to which different compounds are added, such as vitamin E (as DL-α-tocopherol) in AS03 [56] and surfactants in MF59 [57]. Some particularities of these two adjuvants and in which vaccines they are included are discussed below.

•MF59:

MF59 was first approved in Europe in 1997 for use in an influenza vaccine for people over 65 years of age (Fluad^®^) [58], with a clear increase in antibody production compared with standard dose vaccines [17]. Tests carried out during the last decade have shown that this adjuvant allows serological non-inferiority compared to previous influenza vaccines [59], in addition to greater efficacy when its use is extended in this age group [60]. Additionally, the use of this type of influenza vaccine is useful due to its ability to generate heterotypic responses, not only in elderly people [21,37,52,53], but also in children [61].

AS03:

The AS03 adjuvant is also used, like MF59, mainly in influenza vaccines (Fluarix). The origin of the use of this adjuvant is complex, since it began to be explored just before the appearance of the 2009 pandemic flu, which prevented its efficacy from being fully evaluated due to the massive circulation of the A(H1N1)pdm09 virus during this period [62]. Previous data that had been obtained testing this adjuvant against the avian influenza A(H5N1) subtype were promising [63,64]. Another study observed clear benefits against the A(H3N2) subtype in subsequent ad hoc studies, reducing infection by this subtype, as well as mortality from pneumonia and from all causes [65].

▪
Combined and TLR agonist-type adjuvants:


TLR agonist-type adjuvants are systems that stimulate Toll-Like Receptors (TLR) to induce a more potent immune response. TLRs are a family of cellular receptors from the PRRs (Pattern Recognition Receptors) family, which allow the detection of PAMPs (Pathogen-Associated Molecular Patterns) in microorganisms, such as lipopolysaccharide and nucleic acids, in such a way that the immune system can be activated [66,67]. These TLRs are transmembrane receptors that are expressed predominantly in the cells of the innate immune system, some being surface (TLR1, TLR2, TLR4, TLR5 and TLR6) and others intracellular (TLR3, TLR7, TLR8 and TLR9) [67]. TLR agonists are capable of inducing a potent Th2-dependent antibody response [68], as well as, for example, responses at the mucosal level dependent on IgA production [69]. Additionally, mRNA and adenovirus vector vaccines act as their own adjuvants using a TLR approximation. The genetic material that transports both kind of vaccines acts as an activator of TLR for innate immunity stimulation [67].

Sometimes, some adjuvants need to be combined to gain a specific effect. Some adjuvants are, for example, combined with aluminum salts, to enhance their effectiveness in inducing a potent vaccine response. There are currently diverse adjuvant formulations combined with TLR agonists available, some of which are discussed below.

AS04:

AS04 adjuvant is a combined adjuvant that combines the agonist TLR4 (MPL; monophosphoryl lipid A) together with aluminum hydroxide or aluminum phosphate, as appropriate [70]. It is present in vaccines such as Cervarix (against HPV) [71] and in recombinant vaccines against Hepatitis B (such as Fendrix). In the case of Cervarix, for example, it has been shown that the addition of these two adjuvants together produces potent responses against different HPV strains [72], which allows greater protection to be achieved with a single vaccine than if other adjuvants were used, or none. The stimulus generated by this adjuvant at the level of the innate immune system translates into a transient production of cytokines that increase the number of antigen presenting cells (APCs) loaded with the vaccine HPV antigen, giving rise to a much greater activation of T cells in lymph nodes [70].

AS01B:

AS01B is an adjuvant that combines MPL and the saponin QS21 extracted from the bark of the Quillaja saponaria tree, a tree native to Chile. These two compounds form a cholesterol-stabilized liposome [73]. The AS01B system induces a larger antigen-specific cTfh cellular response, which appears to be related to high IgG concentrations and increased numbers of B lymphocytes [74]. The AS01B adjuvant is used, for example, in the vaccine against Herpes zoster (Shyngrix), although there are other novel designs, such as in the vaccine against malaria [74,75].

CpG-ODN:

CpG-ODN is an adjuvant composed of synthetic oligodeoxynucleotides containing unmethylated CpG motifs, capable of activating cells containing type 9 TLRs, with the aim of mounting innate responses characterized by the production of proinflammatory cytokines and Th1 responses [76]. This type of adjuvant improves the function of antigen-presenting cells and enhance both humoral and cellular responses. Some of the vaccines that have tested this type of adjuvant are, for example, vaccines against hepatitis B, anthrax, leishmania, influenza and tetanus, among others [77,78].

Matrix-M:

The Matrix-M™ adjuvant is derived from saponins, natural compounds in the bark of the Quillaja saponaria tree, which stimulate both humoral and cellular responses [79,80]. This compound is an ISCOM (Immune Stimulating Complex) in which saponin mixes with certain fatty acids (cholesterol and phospholipids) to form spherical honeycomb-shaped structures approximately 40 nm in size. The Matrix-M™ adjuvant consists of two different populations of nanoparticles mixed in a defined ratio, 85% Matrix-A and 15% Matrix-C [30,81]. Finally, Matrix-M™ is combined with the antigen of interest to form the final product in the vaccine.

Currently, Matrix-M is used in some vaccines under development, such as the R21/Matrix-M malaria vaccine [80], and some influenza vaccines [79]. In this series of works it has been observed that Matrix-M is especially efficient in the activation of some cells of the innate immune system such as neutrophils, dendritic cells and macrophages, activating both the Th1 and Th2 responses.

The existence of the multitude of widely known adjuvants used in vaccines with a long historical trajectory, in addition to many others not mentioned in this review, allow us to think about their use in future vaccines against COVID-19, in order to achieve a greater immunogenic response of these vaccines, as well as greater logistical capacity for their use, among other factors.

## 4. Adjuvants for New Vaccine Platforms

Despite the negative impact of the pandemic on human health, the need to develop vaccines against COVID-19 in record time has encouraged technologies that were already being explored, such as mRNA or viral vectors, to take the definitive leap towards their commercialization and mass use as a method of preventing infectious diseases. This occurred so much so that barely two years after the appearance of the first mRNA vaccines against COVID-19, there are pharmaceutical industries that are already developing vaccines with this technology against influenza, RSV, or even combined vaccines that mix several respiratory viruses at the same time.

New generation vaccines act as their own adjuvant because they incorporate components capable of being recognized by the immune system as “pathogen-associated molecular patterns” or PAMPs, which trigger cellular recognition patterns leading to the activation of various immune system signaling cascades [82]. In the case of mRNA vaccines, the genetic material acts as one of these PAMPs, being recognized by TLR3, TLR7, TLR8, RIG-I and MDA-5 [83]. However, genetic material does not only have an adjuvant capacity. For example, the lipids that are part of the nanoparticle in which the mRNA is transported (LNPs) have the ability to induce the production of IL-6, which enhances the CD4+ follicular helper T cell (TFH cell) and type B germ cell response [84]. On the other hand, the addition of synthetic nucleotides to the mRNA, in addition to stimulating the production of a greater number of antigens, also enhances the stimulation of different parts of the immune system [85]. In fact, the mRNA for this vaccine is modified in different ways, not only including synthetic nucleotides but also making secondary structures in the 5′ and 3′ untranslated regions for better ribosome accessibility, which has a direct impact on the mRNA lifetime in the cytosol of the cell, and also avoids destruction of the mRNA [86]. Another possible approach for this vaccine is to include dsRNA, which can be recognized by TLR3, MDA-5 and NLRP3, and induce proinflammatory cytokines activating innate and adaptive immunity. This activates myeloid dendritic cells that present the antigen to the naïve T cells through MHC-I molecules [87], thus stimulating CD8+ cells.

In turn, adenoviral vector vaccines, which incorporate the SARS-CoV-2 double-stranded S protein gene (dsDNA) into the genome of the vector itself, activate TLR9 and cGAS [83]. This stimulates the production of interferon and other cytokines that help recruit cells to the puncture site [88], which promotes a proinflammatory state that activates the immune system so that stimulation of the adaptive system is more efficient.

In turn, adenoviral vector vaccines, which incorporate the SARS-CoV-2 double-stranded protein S gene (dsDNA) into the genome of the vector itself, activate TLR9 and cGAS [83]. This stimulates the production of interferon and other cytokines that help recruit cells to the puncture site [88], which promotes a proinflammatory state that activates the immune system so that stimulation of the adaptive system is more efficient.

## 5. Current Adjuvanted COVID-19 Vaccines, Why They Are Useful and Main Challenges

For a vaccine against COVID-19 to be useful, it must face the challenge of the continuous variability of SARS-CoV-2. This implies, in the first instance, that the technology used to produce it allows the vaccine to be adapted and updated with some frequency to include new variants that are emerging. However, and at the same time, it is also desirable not to have to resort to this type of modification as far as possible due to the logistical challenges involved, but rather for the vaccine itself to generate as broad a response as possible to protect against various variants at the same time.

One of the main problems in developing vaccines against RNA-type viruses is their antigenic drift potential. This is the result of several factors. The first is that the RNA molecule is more unstable than the DNA molecule and, therefore, the mutation rate is higher, resulting in their constant and continuous appearance [89]. Additionally, some viruses also lack in their polymerases the ability to correct the errors introduced when duplicating their genetic material, which means that these introduced mutations cannot be eliminated [89]. These two mechanisms together are responsible for the great genetic and antigenic drift of certain viruses such as HIV, influenza, Respiratory Syncytial Virus, and of course SARS-CoV-2. However, not all viruses of this type have the same mutation rate. Of those mentioned, influenza and HIV present the greatest variability, which is up to one logarithm higher than in the case of SARS-CoV-2 [90]. This is because the latter does have a 3’-exonuclease function, which allows the correction of errors introduced by RNA polymerase. Even so, and the pandemic has shown us precisely this, in that the continuous appearance of variants of COVID-19 has been a constant, so, as Leigh Van Valen argued with his Red Queen hypothesis [91], vaccines must run as fast as the virus to stay in the same place. That is, we require the continuous updating of variants to maintain an acceptable immunity against the virus.

As previously mentioned, traditional vaccines are less immunogenic than new vaccine platforms such as mRNA and vector-based vaccines. However, previous designs based on proteins and subunits are still interesting and useful for protecting people, not only against traditionally vaccine-preventable diseases, but also against COVID-19. To stay ahead of the virus and to have a wide arsenal of vaccines, it is necessary to explore all the possibilities at our reach. For that reason, we need to include adjuvants to previous vaccine designs to increase their immunogenicity and usefulness until they reach the one reached by the new designs.

The use of adjuvants in COVID-19 vaccines makes it possible to induce heterotypic responses against different variants or strains of the same virus [92]. Some recent works have shown that some adjuvants can elicit broader responses in animal models [93]. This is a very interesting attribute since it allows a greater durability of the useful life of a vaccine without the need to be updated, taking into account that emergence of new variants will probably be common in the next years after pandemic. However, it is necessary to specify how to select the appropriate adjuvants for these types of vaccines. Furthermore, in this regard, this depends mainly on the characteristics of the vaccine technology. For example, in the case of traditional vaccines it depends mainly on the characteristics of the infection that it is wanted to avoid, so, for example, intracellular pathogens may need adjuvants that trigger CD8+ responses, while others that target the intercellular space may need to stimulate innate immunity, B cells and also CD4+ cells. For increasingly newer technologies, more sophisticated adjuvants that act as PAMPs in a similar way to genetic material or are carriers that activate the immune system are needed.

One of the main interesting points in using adjuvants with COVID-19 vaccines is not only the ability to increase the traditional immune responses presented by classical vaccines, but also the triggering of more sophisticated responses based on innate immunity and also a combination of humoral and cellular response [94]. As some new designs such as mRNA and vector-based vaccines are able to induce wide responses of attraction of cells to the site of injection and also activation of CD4, CD8 and B lymphocytes, adjuvants may play a crucial role in mimicking this behavior in classical vaccines.

At the beginning of 2023, there were a total of 176 vaccine designs at the clinical phase and 199 at the pre-clinical phase against COVID-19 [2]. Most of these vaccines are inactivated and are designed with protein subunits [95]. Of those that are in the clinical phase, 33 have some type of adjuvant, including 21 vaccines with protein subunits, 5 vaccines with virus-like particles (VLP), 5 vaccines with inactivated whole viruses, 1 vaccine with non-replicating viral vectors, 1 vaccine with replicating viral vectors, and 1 vaccine with COVID-19 in addition to another microorganism, such as influenza. Some of the most advanced designs, because they are in the review phase or because they are already authorized for use, are recombinant protein vaccines that use some type of adjuvant in their formulation. Table 1 shows the main characteristics of these vaccines.

Additionally, there are other vaccines against COVID-19 that contain adjuvants and that are currently approved by the EMA (European Medicines Agency), such as VLA2001 (Valneva Austria GmbH), which is an inactivated complete virus vaccine that uses the dual adjuvant aluminum hydroxide-CpG1018 [98,99]. Other vaccines that contain adjuvants are approved in other countries, such as CoronaVac (Sinovac Life Sciences Co., Ltd.), which is an inactivated vaccine that uses aluminum hydroxide as an adjuvant [100], or COVILO (Sinopharm), which contains aluminum hydroxide [101].

The main challenge of adjuvants is that currently authorized adjuvants present some gaps in the immunization process that make them incomplete in terms of overall response. Each adjuvant has a limited stimulation system, which only allows for the “touching” of some parts of the complex immune response. Therefore, one of the possible solutions could be the use of combined adjuvants, which can stimulate different parts of the immune response at the same time, and also boost the response jointly [102]. However, these combinations should be made by searching for the best optimal ratio and the best associated adjuvants, thus looking for the combination that provides an additive and truly synergistic effect. Another important challenge for adjuvants is that some of them can elicit strong inflammatory responses that can interfere with antigen expression, negatively impacting immunogenicity. One example is the LNP stimulation. However, there are some countermeasures to solve this problem, such as partial substitution of 10% of cholesterol in the LNP formulation by dexamethasone, which decreases TNFα production and allows for better antigen expression when mRNA is administered [103].

## 6. Final Arguments

The portfolio of vaccines that explore different manufacturing systems and that include one or more adjuvants is extensive and seems an interesting bet for future vaccines against COVID-19. Currently they represent approximately 20% of the vaccines that are in the phase of clinical evaluation, but it is likely that there will be new designs that jump from the pre-clinical phase to the clinical one in the coming months or years, increasing the number of available vaccines against this new disease. The main advantage of using adjuvants is that they can be applied to most of the classical vaccine designs available prior to the pandemic, so are a way to increase the usefulness of these kinds of vaccines to complete the diversity of models with new vaccines such as mRNA and vector-based vaccines.

Adjuvants are especially important for their use in vaccines against COVID-19 due to the special damage that this new disease has caused in the elderly and those with conditions that affect their ability to fight infectious diseases. Adjuvanted vaccines have traditionally shown an increased response to vaccination in these types of risk groups, which has helped to achieve levels of protection that in many cases resemble those achieved in healthy young adults. This is important to protect the most vulnerable, since, in addition, this type of vaccine implies a more lasting response, which has a positive impact on the prevention of infectious diseases in the medium and long term.

## Figures and Tables

**Figure 1 vaccines-11-00902-f001:**
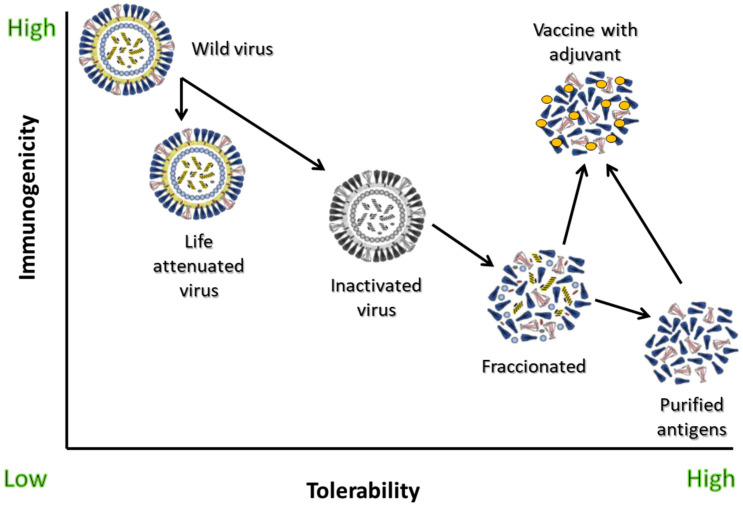
Immunogenicity and tolerability of different types of influenza vaccines and of wild-type virus challenge. Adapted from [8].

**Table 1 vaccines-11-00902-t001:** Characteristics of adjuvanted vaccines against COVID-19 [2,96,97].

Name of Vaccine	Developer	Type of Vaccine	Type of Adjuvant	Administration Route	Status EU/Clinical Phase	Indications EU
Nuvaxovid (NVX-CoV2373)	Novavax	Protein subunit	MATRIX-M	IM	Authorized	Primary vaccination and booster
VidPrevtyn beta	Sanofi-GSK	Protein subunit	AS03	IM	Authorized	Booster dose
PHH-1V	HIPRA	Protein subunit	MF59	IM	Authorized	Booster dose
VLA2001	Valneva	Whole inactivated virus	CpG 1018	IM	Authorized	Primary vaccination and booster
VXA-CoV2-1	Vaxart	Viral vector (non replicative)	TLR3 stimulator	Oral	2	NA
SCB-2019	Clover Biopharmaceuticals Inc./Dynavax	Protein subunit	CpG 1018/Alum Hydroxide	IM	3	NA
Spikogen	Vaxine Pty Ltd./CinnaGen Co.	Protein subunit	CpG55.2	IM	4	NA
V451_07	CSL Ltd. + Seqirus + University of Queensland	Protein subunit	MF59	IM	2/3	NA
MVC-COV1901	Medigen	Protein subunit	CpG 1018	IM	4	NA
FINLAY-FR1 (Soberana 01)	Instituto Finlay de Vacunas	Protein subunit	Alum hydroxide	IM	2	NA
FINLAY-FR-2 (Soberana 02)	Instituto Finlay de Vacunas	Protein subunit	Alum hydroxide	IM	3	NA
Covifenz	Medicago Inc.	VLP	AS03	IM	4	NA
Nanocovax	Nanogen Pharmaceutical Biotechnology	Protein subunit	Alum hydroxide	IM	3	NA
COVAC-1 & COVAC-2	University of Saskatchewan	Protein subunit	Alum hydroxide	IM	2	NA
GBP510	SK Bioscience Co., Ltd. and CEPI	Protein subunit	Alum hydroxide	IM	3	NA
VBI-2902a	VBI Vaccines Inc.	VLP	Alum phosphate	IM	1/2	NA
EuCorVac-19	POP Biotechnologies and EuBiologics Co., Ltd.	Protein subunit	Alum hydroxide	IM	3	NA
Koçak-19	Kocak Farma	Whole inactivated virus	Alum hydroxide	IM	1	NA
COVIVAC	Institute of Vaccines and Medical Biologicals, Vietnam	Viral vector (replicative)	CpG 1018	IM	1/2	NA
ABNCoV2	Radboud University	VLP	MF59	IM	3	NA
Tübitak	TÜBITAK	Whole inactivated virus	CpG 1018/Alum Hydroxide	SC	1	NA
VAX1	Baiya Phytopharm Co., Ltd.	Protein subunit	Alhydrogel	IM	1	NA
202-CoV	Shanghai Zerun Biotechnology	Protein subunit	CpG7909	IM	1/2	NA
Novavax COVID-19 Omicron	Novavax	Protein subunit	Matix-M1	IM	3	NA
PIKA	Yisheng Biopharma	Protein subunit	polyl:C	IM	2/3	NA
LYB001	Yantai Patronus Biotech Co., Ltd.	VLP	Alum hydroxide	IM	3	NA
Indovac	PT Bio Farma	Protein subunit	CpG 1018/Alum Hydroxide	IM	3	NA
Vaccine against Beta	University of Melbourne	Protein subunit	MF59	IM	1	NA
CIC (SARS-CoV-2 + Influenza)	Novavax	Protein subunit	Matix-M1	IM	1/2	NA
VBI-2901e	VBI Vaccines Inc.	VLP	Alum phosphate/E6020	IM	1	NA
CoronaVac	Sinovac	Whole inactivated virus	Alum hydroxide	IM	4	NA
Covilo	Sinopharm	Whole inactivated virus	Alum hydroxide	IM	4	NA
Zifivax	Anhui Zhifei Longcom	Protein subunit	Alum hydroxide	IM	4	NA

Note: IM, Intramuscular; SC, Subcutaneous; NA, Non-applicant.

## Data Availability

Not applicable.

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
