# Peer review of "Adjuvants for COVID-19 Vaccines"

_vaccines, 2023, doi:10.3390/vaccines11050902_

Round 1

Reviewer 1 Report

In the manuscript submitted to Vaccines (id nº vaccines-2332965), entitled “Adjuvants for COVID-19 vaccines”, by Castrodeza-Sanz et al. the authors propose to “analyze the advantages and disadvantages of adjuvants, as well as their use in current and future vaccines against COVID-19”.

Overall, the manuscript reads well. However, some English polishing can be done. Additionally, although the review on adjuvants is quite interesting, the discussion of the adjuvanted COVID vaccines developed is limited. Please check below a few specific comments for you to have a change to improve your manuscript (should you agree with them).

Major points:

1-     With respect to the sub-heading “They stimulate the immune response beyond the humoral response”, I think you cannot generalize. You either need to clearly highlight that this may be the case for some adjuvants. Additionally, we cannot also forget that the type of antigen matters when we are thinking of T cell responses, as does the antigen dose. Therefore, if you want to mention this, you need to re-write this section to accommodate the notions above.

2-     In the list of COVID vaccines, you only include the 3 that are (almost) approved for use. However, you mention 32 in the clinical phase that have some type of adjuvant. I would list all of them in this table. Otherwise, the information is quite abstract and limited.

Minor points:

1-     The use of English language should be polished. Please check a few examples

a.      Line 17 should read “…much better tolerability profiles…” (either use much better profiles, or a better profile, for accuracy).

b.      Line 110 should read “…through, for example…”

c.      Line 137 should read “…to achieve optimal response.”

d.      Line 188 should read “One of the main problems of some antigens is that they are soluble…”

e.      Line 298 should read “Sometimes, some adjuvants need…”

f.       Line 300 should read “There are currently diverse adjuvant formulations combined with TLR agonists…”

g.      Line 353: please consider revising to “Current adjuvanted COVID-19 vaccines and they are useful”

h.      Line 362: please consider revising to “…RNA-type viruses is their antigenic drift potential.”

i.       Line 375 should read “…with his Red Queen hypothesis…”

Author Response

In the manuscript submitted to Vaccines (id nº vaccines-2332965), entitled “Adjuvants for COVID-19 vaccines”, by Castrodeza-Sanz et al. the authors propose to “analyze the advantages and disadvantages of adjuvants, as well as their use in current and future vaccines against COVID-19”.

Overall, the manuscript reads well. However, some English polishing can be done. Additionally, although the review on adjuvants is quite interesting, the discussion of the adjuvanted COVID vaccines developed is limited. Please check below a few specific comments for you to have a change to improve your manuscript (should you agree with them).

Major points:

  • With respect to the sub-heading “They stimulate the immune response beyond the humoral response”, I think you cannot generalize. You either need to clearly highlight that this may be the case for some adjuvants. Additionally, we cannot also forget that the type of antigen matters when we are thinking of T cell responses, as does the antigen dose. Therefore, if you want to mention this, you need to re-write this section to accommodate the notions above
  •  
  • Authors: We have written this paragraph further in order to explain more in detail the issues that you suggested.

2-     In the list of COVID vaccines, you only include the 3 that are (almost) approved for use. However, you mention 32 in the clinical phase that have some type of adjuvant. I would list all of them in this table. Otherwise, the information is quite abstract and limited.

Authors: We have updated the table 1 and included all the vaccines from different resources.

Minor points:

1-     The use of English language should be polished. Please check a few examples

  1. Line 17 should read “…much better tolerability profiles…” (either use much better profiles, or a better profile, for accuracy).

Authors:  Changed.

  1. Line 110 should read “…through, for example…”

Authors:  Changed

  1. Line 137 should read “…to achieve optimal response.”

Authors: Changed

  1. Line 188 should read “One of the main problems of some antigens is that they are soluble…”

Authors: Changed

  1. Line 298 should read “Sometimes, some adjuvants need…”

Authors: Changed

  1. Line 300 should read “There are currently diverse adjuvant formulations combined with TLR agonists…”

Authors: Changed

  1. Line 353: please consider revising to “Current adjuvanted COVID-19 vaccines and they are useful”

Authors:  Changed

  1. Line 362: please consider revising to “…RNA-type viruses is their antigenic drift potential.”

Authors: Changed

  1. Line 375 should read “…with his Red Queen hypothesis…”

Authors: Changed

Reviewer 2 Report

The authors wrote a comprehensive review on the adjuvants used in COVID-19 vaccines. As they mentioned, COVID-19 pandemic accelerated the development of new vaccination systems, including the application of novel adjuvants. Therefore, this review is very valuable for the deeper understanding of new technologies on adjuvants used in viral vaccines.

Here are some questions and suggestions:

1. Inactivated COVID-19 vaccines are mainly used in very large population. In some countries, they are the major form of vaccine used against COVID-19. Some people believed that the insufficient protective capacity of inactivated vaccines compared with mRNA vaccines can be attributed to the traditional adjuvants (aluminum hydroxide) used in inactivated vaccines . Hopefully the authors can discuss more about the roles of adjuvants in inactivated vaccines.

2. I suggested the authors add ZF2001, a protein subunit COVID-19 vaccine (N Engl J Med 2022; 386:2097-2111 DOI: 10.1056/NEJMoa2202261), in their Table 1. This vaccine has been widely used in China. 

Minor points:

Line 60: ...became more sophisticated, ... This expression is confusing. Actually the  immunostimulant capacity of recent vaccines decreased because the immunogens became more simplified and purified in bio-engineered vaccines.

Author Response

The authors wrote a comprehensive review on the adjuvants used in COVID-19 vaccines. As they mentioned, COVID-19 pandemic accelerated the development of new vaccination systems, including the application of novel adjuvants. Therefore, this review is very valuable for the deeper understanding of new technologies on adjuvants used in viral vaccines.

Here are some questions and suggestions:

  1. Inactivated COVID-19 vaccines are mainly used in very large population. In some countries, they are the major form of vaccine used against COVID-19. Some people believed that the insufficient protective capacity of inactivated vaccines compared with mRNA vaccines can be attributed to the traditional adjuvants (aluminum hydroxide) used in inactivated vaccines . Hopefully the authors can discuss more about the roles of adjuvants in inactivated vaccines.

Authors: We included a new paragraph after the Figure 1 explaining your suggestions.

  1. I suggested the authors add ZF2001, a protein subunit COVID-19 vaccine (N Engl J Med 2022; 386:2097-2111 DOI: 10.1056/NEJMoa2202261), in their Table 1. This vaccine has been widely used in China. 

Authors: We renewed the table 1 including all vaccines that are in any clinical phase and included the vaccine that you suggested and the reference.

Minor points:

Line 60: “...became more sophisticated, ...” This expression is confusing. Actually the  immunostimulant capacity of recent vaccines decreased because the immunogens became more simplified and purified in bio-engineered vaccines.

Authors: We agree with your comment, it is difficult for the reader to understand what we try to mean with “sophisticated”. For that, we change it by what you mentioned, that the antigens included in this kind of vaccines are simpler and more purified.

Reviewer 3 Report

The review titled “Adjuvants for COVID-19 vaccines’ by Castrodeza-Sanz et al explains about the different type of adjuvants used for the development of vaccines. This review is well written, however, still authors need to explain few important points before it gets accepted. Overall, this review can be accepted after considering major revision.    

11. Authors could also mention what type of adjuvants can be potentially used for mRNA vaccines. Also, is it possible to use dsRNA as an adjuvant? Please explain.

22. How to select specific adjuvant for traditional, DNA, peptide or mRNA vaccines? 

33. What type of immune response is typically observed for different type of adjuvants? Please add information with relevant figure(s).

44. Is there any dose response for adjuvants? Will higher the adjuvant dose have any effect for protein translation in mRNA vaccine? Please explain.

55. Page 2, Paragraph 2. Please add references. For ex. “Many of the older inactivated vaccines were made up of the whole organism, which offered a moderate vaccine response (immunogenicity)…” and “To avoid this reactogenicity, vaccines progressively evolved to designs that contained only parts of the microorganism, subunits, or even purified antigens..”

66. Page 4, add references. “….they require multiple doses to achieve optimal”.

77. Page 3, add relevant references. For section. “They make it possible to reduce the amount of antigen needed for each vaccine dose”

88. Also, Its not clearly understand that how authors justify about reducing the antigen in vaccines. For ex. “They make it possible to reduce the amount of antigen needed for each vaccine dose”.  Please explain.

99. Authors need to focus on this section “Current adjuvanted COVID-19 vaccines and why to use on them”. Authors touched few points about other vaccines apart from COVID 19, it’s better to add more relevant information about other vaccines as well.

110.  Also, please add what are the potential challenges while using adjuvants.

Author Response

The review titled “Adjuvants for COVID-19 vaccines’ by Castrodeza-Sanz et al explains about the different type of adjuvants used for the development of vaccines. This review is well written, however, still authors need to explain few important points before it gets accepted. Overall, this review can be accepted after considering major revision.    

  1. Authors could also mention what type of adjuvants can be potentially used for mRNA vaccines. Also, is it possible to use dsRNA as an adjuvant? Please explain.

Authors: We have included a new subheading explaining who work these kind of vaccines and how acts as their own-adjuvants, including also information on how dsRNA works.

  1. How to select specific adjuvant for traditional, DNA, peptide or mRNA vaccines? 

Authors: We included information on how to select the appropriate adjuvant depending on the vaccine type in the last subheading.

  1. What type of immune response is typically observed for different type of adjuvants? Please add information with relevant figure(s).

Authors: Thank you for your comment. However, we do not agree to include a specific subheading with more detailed information about the immune response that each type of adjuvant triggers since this information was included along different parts of the manuscript and we think that it is not necessary to be included because this is not the aim of this review.

  1. Is there any dose response for adjuvants? Will higher the adjuvant dose have any effect for protein translation in mRNA vaccine? Please explain.

Authors: Yes, indeed, the modifications in the mRNA (synthetic nucleotides, secondary structures, etc.) may modify the expression of the antigen protein, so it impacts the number of protein copies that will be delivered by the immune system. This clearly impact the number of antigens being able to be recognized by the immune system, so the adjuvant dose clearly impacts the immunization in the case of mRNA vaccines. We included information about that in the subheading 4.

  1. Page 2, Paragraph 2. Please add references. For ex. “Many of the older inactivated vaccines were made up of the whole organism, which offered a moderate vaccine response (immunogenicity)…” and “To avoid this reactogenicity, vaccines progressively evolved to designs that contained only parts of the microorganism, subunits, or even purified antigens..”

Authors: For the first phrase, the reference 7 was included. For the second phrase, it was included the reference number 8.

  1. Page 4, add references. “….they require multiple doses to achieve optimal”.

Authors: Included

  1. Page 3, add relevant references. For section. “They make it possible to reduce the amount of antigen needed for each vaccine dose”

Authors: There were just 3 references (9, 21 and 23) included that talk exactly about this issue.

  1. Also, Its not clearly understand that how authors justify about reducing the antigen in vaccines. For ex. “They make it possible to reduce the amount of antigen needed for each vaccine dose”.  Please explain.

AAuthors:We think that this issue is well explained in this paragraph. Our assumption relies in that the less need of an adjuvant let to make more vaccines doses, and for that reason you can vaccine more people in less time. This is so important in a pandemic or other situations when you need to vaccinate a high number of persons.

  1. Authors need to focus on this section “Current adjuvanted COVID-19 vaccines and why to use on them”. Authors touched few points about other vaccines apart from COVID 19, it’s better to add more relevant information about other vaccines as well.ç

Authors: We updated the table 1 including all the vaccine designs that have adjuvants. Also we included more relevant information about the use of appropriate adjuvants in different vaccines, and also the main challengues of adjuvants regarding your next suggestion.

  1. Also, please add what are the potential challenges while using adjuvants.

Authors: We included a paragraph at the end of the last subheading talking about this issue.

Round 2

Reviewer 1 Report

My concerns were properly addressed. Do note that the text additions in red need english language editing.

Author Response

Thank you for your appreciations. We made an extensively English editing in the parts that you specified.

Reviewer 3 Report

The review titled “Adjuvants for COVID-19 vaccines’ by Castrodeza-Sanz et al explains about the different type of adjuvants used for the development of vaccines. Authors addressed all questions along with the incorporation of relevant information in the manuscript.

However, a small correction in page 11, line 498. “…replacement of cholesterol in LNP formulation with dexamethasone…”. It’s a partial substitution, not a complete replacement. Please modify accordingly (no need to review further). Apart from that I agree to accept this manuscript for publication.       

Author Response

Thank you for your comment. We added to this paragraph that the replacement of cholesterol by dexamethasone in around 10%.